# Genetic hierarchy and temporal variegation in the clonal history of acute myeloid leukaemia

Pierre Hirsch[1,2,3,4,*], Yanyan Zhang[5,6,*], Ruoping Tang[4], Virginie Joulin[5,6], Hélène Boutroux[1,2,3,7], Elodie Pronier[6], Hannah Moatti[1,2,3], Pascale Flandrin[1,2,3], Christophe Marzac[8], Dominique Bories[9], Fanny Fava[1], Hayat Mokrani[6], Aline Betems[6], Florence Lorre[10], Rémi Favier[8], Frédéric Féger[8], Mohamad Mohty[1,3], Luc Douay[1,2,8], Ollivier Legrand[1,2,3,4], Chrystèle Bilhou-Nabera[1,2,3,8], Fawzia Louache[5,6,**] & François Delhommeau[1,2,3,8,**]

In acute myeloid leukaemia (AML) initiating pre-leukaemic lesions can be identified through three major hallmarks: their early occurrence in the clone, their persistence at relapse and their ability to initiate multilineage haematopoietic repopulation and leukaemia *in vivo*. Here we analyse the clonal composition of a series of AML through these characteristics. We find that not only *DNMT3A* mutations, but also *TET2*, *ASXL1* mutations, core-binding factor and *MLL* translocations, as well as del(20q) mostly fulfil these criteria. When not eradicated by AML treatments, pre-leukaemic cells with these lesions can re-initiate the leukaemic process at various stages until relapse, with a time-dependent increase in clonal variegation. Based on the nature, order and association of lesions, we delineate recurrent genetic hierarchies of AML. Our data indicate that first lesions, variegation and treatment selection pressure govern the expansion and adaptive behaviour of the malignant clone, shaping AML in a time-dependent manner.

[1] Sorbonne Universités, UPMC Univ Paris 06, UMR_S 938, CDR Saint-Antoine, F-75012 Paris, France. [2] INSERM, UMR_S 938, CDR Saint-Antoine, F-75012 Paris, France. [3] Sorbonne Universités, UPMC Univ Paris 06, GRC n°7, Groupe de Recherche Clinique sur les Myéloprolifération Aiguës et Chroniques MYPAC, F-75012 Paris, France. [4] AP-HP, Hôpital St Antoine, Service d'Hématologie clinique et de thérapie cellulaire, F-75012 Paris, France. [5] Institut National de la Santé et de la Recherche Médicale (INSERM), UMRS 1170, CNRS GDR 3697 Micronit, 94805 Villejuif, France. [6] Institut Gustave Roussy, Univ Paris-Sud, Université Paris Saclay, 94805 Villejuif, France. [7] Department of Pediatric Hematology and Oncology, AP-HP, Hôpital Armand-Trousseau, F-75012 Paris, France. [8] AP-HP, Hôpital Saint-Antoine & Hôpital Armand-Trousseau, Service d'hématologie biologique, F-75012 Paris, France. [9] AP-HP, Hôpital Henri Mondor, Unité d'Hématologie Moléculaire, F-94010 Créteil, France. [10] AP-HP, Hôpital Saint-Antoine, Laboratoire commun de biologie et génétique moléculaires, F-75012 Paris, France. * These authors contributed equally to this work. ** These authors jointly supervised this work. Correspondence and requests for materials should be addressed to F.D. (email: francois.delhommeau@aphp.fr).

Acute myeloid leukaemia (AML) emerges from haematopoietic stem/progenitor cells (HSPCs) that acquire multiple genomic or chromosomal aberrations. It has been proposed to affiliate these heterogeneous aberrations to eight distinct functional groups according to their known or putative consequences on signalling, chromatin modification, DNA methylation, the cohesin complex, transcription factors, NPM1, the splicing machinery and tumor suppressors[1,2]. Some of these genetic lesions, including DNMT3A, TET2 and ASXL1 mutations, are considered as initiating events and are believed to result in clonal expansion of mutant HSPCs, leading to clonal haematopoiesis of indeterminate potential (CHIP)[3–6], a condition that predisposes to the subsequent acquisition of leukaemic mutations. This model of leukaemogenesis fits with the results from previous studies that investigated the clonal evolution of AML with normal karyotype, showing for instance that DNMT3A mutations precede NPM1 or FLT3 mutations[7–9].

DNMT3A mutations genetically define pre-leukaemic stem cells in mouse xenotransplantation models of AML because they are sufficient to provide to mutant HSPCs a multilineage repopulation potential[8]. However, as DNMT3A mutations account for less than 30% of individuals with CHIP[3] or adult AMLs[2], the genetic variety of these diseases raises the question whether other chromosomal or genetic lesions behave as initiating pre-leukaemic events. In a particular subtype of AML, acute promyelocytic leukaemia, the PML/RARA fusion has been shown to be the initiating lesion[10]. In other types of AMLs, some lesions like TET2 or IDH2 mutations, as well as core-binding factor (CBF) or MLL translocations, are also thought to be initiating events, as they occur early in the clonal history, or can lead to pre-leukaemic haematopoiesis in xenotransplantation models[7,11–14].

In the present work, we examine how the models of initiation and clonal evolution already described in some subtypes of AML could be generalized to most cases. We analyse 74 non-promyelocytic AMLs, including 53 consecutive unselected cases reflecting the genetic variety of AMLs, in order to identify the events that fit with three major hallmarks of pre-leukaemic initiating events: (1) their early occurrence in the clone[7,8,10,11], (2) their persistence at relapse[8,11] and (3) their ability to provide a multilineage selective advantage to mutant over normal HSPCs in vivo[7,8]. By an integrated clonal reconstruction of AML using cytogenetic, molecular, targeted sequencing, whole exome sequencing and single-cell-derived colony genotyping data, we show that DNMT3A mutations, but also TET2, ASXL1, as well as CBF, MLL and chromosome 20q rearrangements are the first driver events in most AMLs, can persist in remission, and are retained when patients experience relapse. Analysis of early and late relapse samples reveals that clones with these stable alterations variegate increasingly with time from diagnosis to recurrence. Moreover, cells from patients with these early lesions can repopulate bone marrow of xenotransplanted NOD/SCID/IL-2Rg$_c$-null (NSG) mice with leukaemic or non-leukaemic engraftment, a functional signature of expanding pre-leukaemic stem cells[8]. Finally, our determination of clonal composition allows us to delineate genetic hierarchies that suggest distinct mechanisms of disease initiation and clonal expansion.

## Results

**Clonal composition of AML.** To identify early putative initiating events, we analysed the clonal composition of 53 consecutive AMLs (Supplementary Fig. 1) by routine cytogenetic and PCR techniques, and targeted resequencing of 122 genes recurrently mutated in AML (Supplementary Tables 1 and 2). Additional whole-exome sequencing was performed in four MLL-rearranged cases where only a few lesions were identified by targeted resequencing, one normal karyotype AML (UPN 2014-018), and one AML with monosomy 7 (UPN 2014-009) (Supplementary Tables 3,4, Supplementary Data 1). As multiple lesions can lead to various functional consequences[1,2], we arbitrarily re-classified them from eight to four categories according to their main putative functional consequences (Supplementary Table 5). The first category comprises lesions disturbing epigenetic regulation: mutations in DNA methylation and chromatin modifiers, MLL and CBF translocations[15–17], and del(20q) (ref. 18). The second category includes mutations in splicing factors, transcription factors, and in NPM1, all impairing haematopoietic differentiation[19–22]. The third category refers to mutations in genes that regulate proliferation and the fourth category to all other events.

Twenty-six out of the 53 samples were seeded in methylcellulose cultures to obtain single-cell-derived colonies. Colonies from 15 out of these 26 AMLs were next analysed by molecular and cytogenetic methods to infer the architecture of the principal clones from the existence of colonies with none, all or combinations of lesions detected in the bulk DNA. The 11 remaining cases were not further analysed due to either a lack of material or an insufficient number of detectable lesions.

We first focused on eight patients with normal karyotype and mutations in TET2, DNMT3A or ASXL1, the most frequently mutated genes in CHIP[3–5]. In five patients, TET2 mutations were found in first position, preceding DNMT3A mutations in three double-mutant cases. In two other patients without TET2 mutations, DNMT3A mutations were found first (Fig. 1a,b, Supplementary Fig. 2). The last patient had an ASXL1 mutation followed by lesions in EZH2, RUNX1 and FLT3 (Fig. 1b, Supplementary Fig. 2). In three cases (UPN2014-001, UPN2014-020, UPN2014-022), we found evidence for branching subclones involving variants in TET2, FLT3 and NRAS, all detected with low allele frequencies in the bulk AML DNA (Supplementary Table 3).

We then analysed the seven AMLs with abnormal karyotypes. Informative polymorphisms and fluorescent in situ hybridization (FISH) were used to detect losses of heterozygosity (LOH) and translocations, respectively. In one case (UPN 2014-015), del(20q) was the founder event, followed by a splicing machinery mutation in U2AF1, and a mutation in the tyrosine kinase RET (Fig. 1c). In a second patient (UPN 2014-008), del(16q) was found after an U2AF1 mutation, and preceding an FLT3-internal tandem duplication (ITD) with LOH (Fig. 1e, Supplementary Fig. 3). In one patient with monosomy 7 (UPN 2014-009), the 122 gene-panel revealed mutations in SF3B1, PTPN11 and FLT3 (Supplementary Table 3). Exome sequencing did not identify any additional candidate initiating variant (Supplementary Table 4, Supplementary Data 1). Single-cell-derived colony analysis showed that the first event was the mutation in SF3B1, followed by the PTPN11 one, the loss of chromosome 7 and the FLT3 mutation. In four other patients (UPN 2014-003, UPN 2013-004, UPN 2014-019 and UPN 2013-001), MLL translocations were identified as first events by FISH, and RAS or FLT3 pathway mutations as last lesions by single-cell-derived colonies (Fig. 1d,e, Supplementary Fig. 3).

Taken together, single-cell-derived colony analyses revealed that mutations in epigenetic regulators, MLL rearrangements, and del(20q) are early events. Conversely, in 12/15 cases, proliferation-associated events, including FLT3 and RAS pathway mutations, were found as last events.

**Early event retention and temporal changes in relapse AML.** A second characteristic of pre-leukaemic lesions is that they persist at relapse[8,11]. To investigate this, we listed the changes in

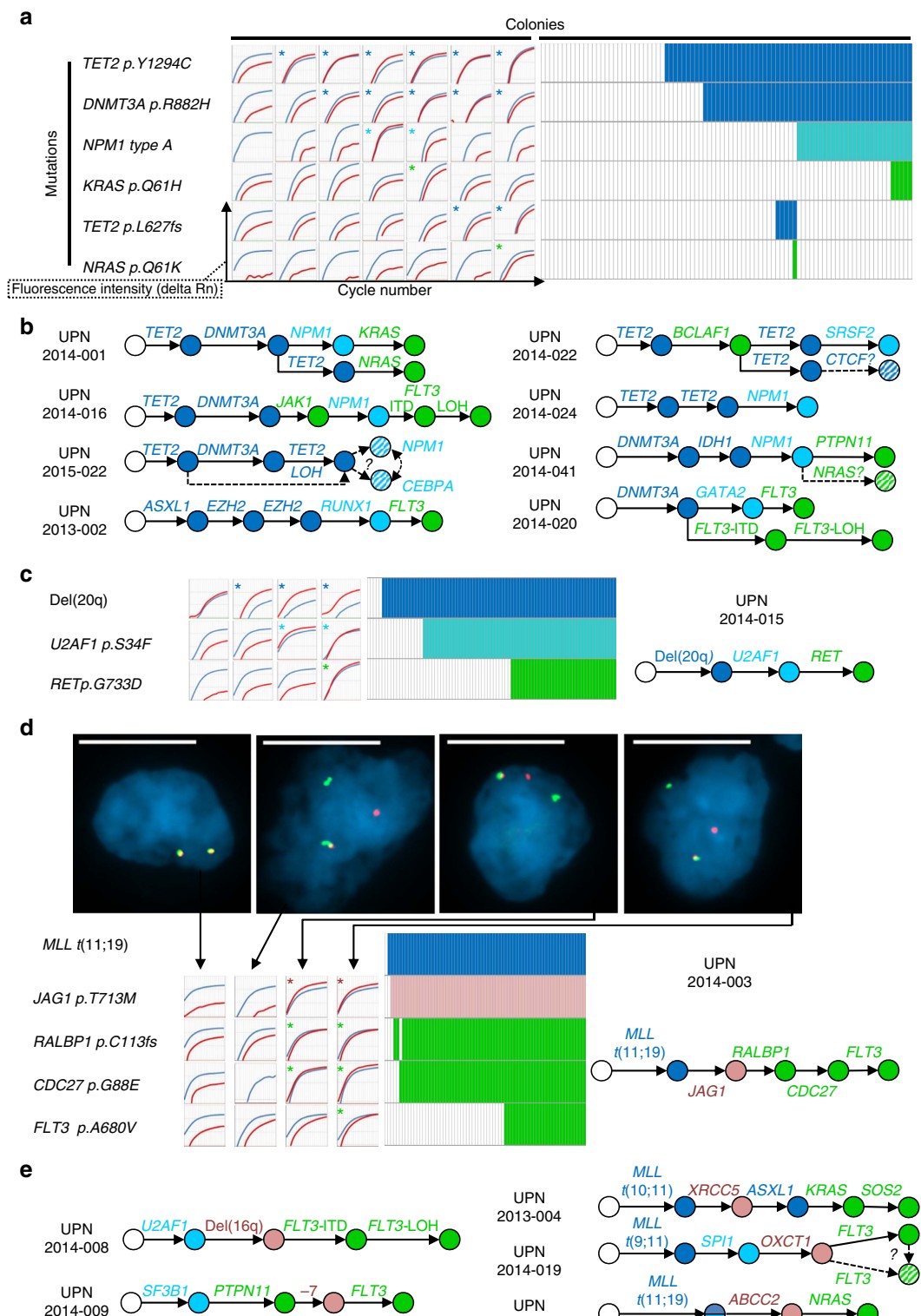

**Figure 1 | Single-cell-derived colony analysis identifies a recurrent order of acquisition of AML mutations and chromosomal rearrangements. (a)** Left panel: allele specific (AS) PCR amplification plots of mutant (red) and wild type (blue) alleles in seven colonies from patient UPN2014-001. Asterisks indicate mutant allele detection. Right panel: results of AS assays (rows) in 87 colonies (columns). Mutant colonies are indicated by coloured boxes. Blue: lesions in epigenetic regulators; cyan: lesions involving *NPM1*, transcription or splicing factors; green: proliferative lesions; pink: other lesions. **(b)** Mutation order in eight patients with normal karyotype AML. Dashed arrows and hatched circles indicate positions that could not be determined. **(c)** Analysis of 85 colonies from patient UPN2014-015 as in **a**. To detect del(20q), an informative single-nucleotide polymorphism (SNP)(rs11556379) was used. The inferred order of lesions is shown. **(d)** Genotyping analyses, as in **a**, and FISH analyses of colonies from patient UPN2014-003. Dissociations of the red and green signals indicate *MLL* rearrangements. Nuclei were stained with 4,6-diamidino-2-phenylindole. Bars indicate 10 µm. Right panel: results of FISH and AS assays, as in **a**. The inferred order of lesions is shown. **(e)** Order of lesions in five additional abnormal karyotype AMLs.

clonal composition between diagnosis and relapse in 22 patients, including five additional patients who had experienced relapse over 5 years after diagnosis (Supplementary Fig. 1). In all patients but two, from one to 20 genomic or chromosomal changes were detected (Fig. 2a, Supplementary Tables 1,6,7, Supplementary Fig. 4, Supplementary Data 2). In contrast to epigenetic modifying events, which were mostly unchanged in the relapse clones (16 changes out of 55 lesions), late proliferative events changed frequently (28 changes out of 36 events, $P < 0.0001$, Fisher's exact test) (Fig. 2b). Moreover, all epigenetic lesions identified as first events in our previous colony analyses —DNMT3A, TET2, ASXL1 mutations, MLL rearrangement and chromosome 20q deletions—were retained at relapse (Fig. 2a, Supplementary Table 7). We also observed that, while the number of retained lesions decreased with time to relapse (mean ± s.e.m.: 4.1 ± 0.6, $n = 17$ before 5 years versus 1.6 ± 0.7, $n = 5$ after 5 years, $P = 0.016$, Mann–Whitney test), the number of variegating lesions increased in the meantime (mean ± s.e.m.: 2.5 ± 0.4, $n = 17$ before 5 years versus 8.0 ± 3.3, $n = 5$ after 5 years $P = 0.023$, Mann–Whitney test) (Fig. 2c,

Supplementary Table 7 and Supplementary Fig. 4). Of note, four out the five patients with late relapses had multiple changes in clonal composition. In one patient (UPN2015-003), we concluded that a second disease, with fully distinct genotype and karyotype from those of the initial leukaemia, had occurred. In the three other patients, persistent DNMT3A mutant clones had lost most mutations, including an NPM1 mutation (UPN2015-014), an IDH2 mutation (UPN2015-004), and a RUNX1 mutation (UPN2015-005). At relapse, these founding clones had re-evolved with several changes including new mutations in NPM1, IDH1 and RUNX1, respectively (Fig. 2a, Supplementary Table 7 and Supplementary Fig. 4). None of the relapse specific mutations was detected by deep sequencing with a 0.001 sensitivity in the diagnosis sample (Supplementary Table 8). In patient UPN2015-014, the analysis of sequential samples showed that the relapse arose after the emergence of an IDH2 mutant sub-clone, which had subsequently acquired a second NPM1 type A mutation identical to that of the diagnosis clone (Fig. 2a, Supplementary Table 8). These results suggest that leukaemia treatments had reset the clones back to their

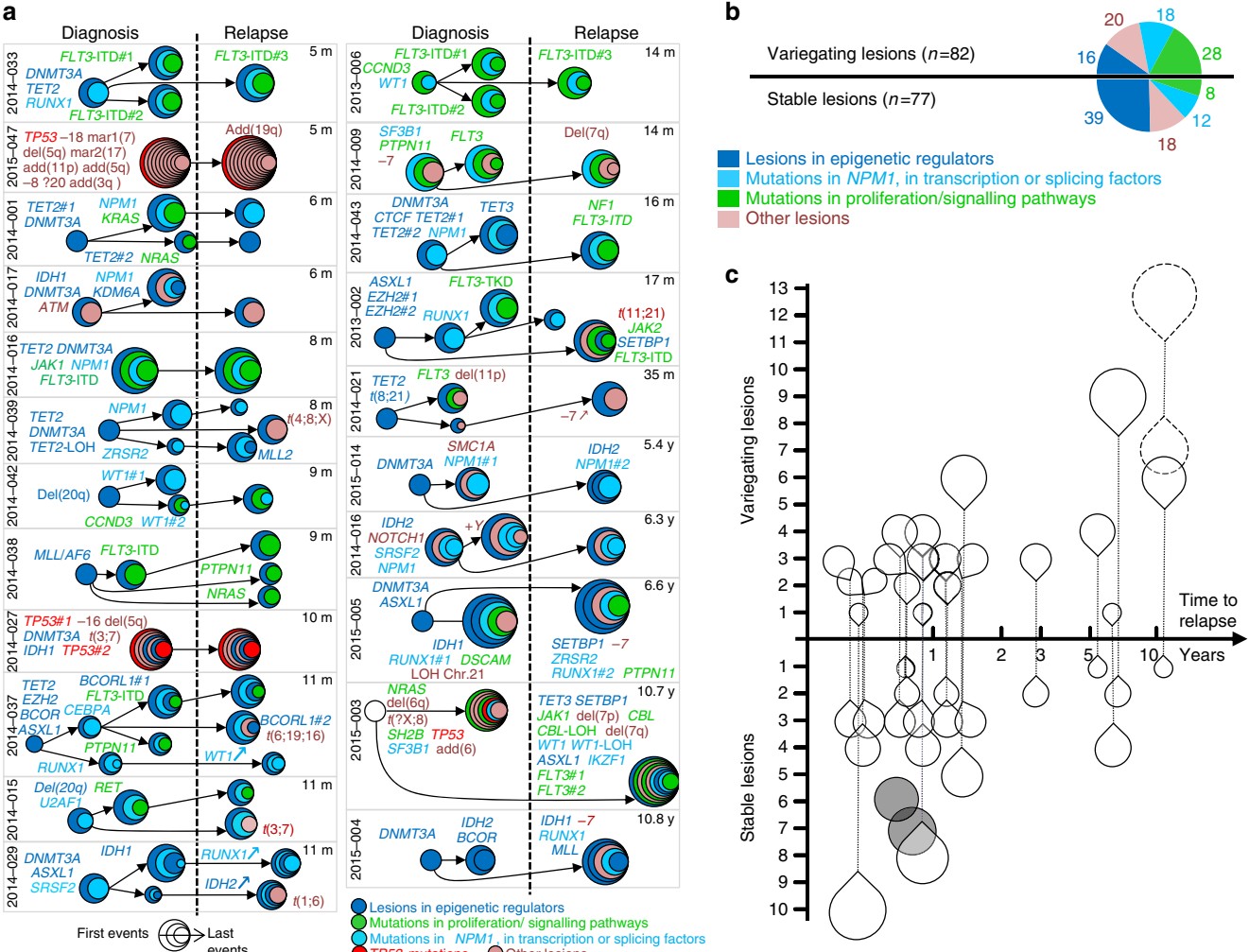

Figure 2 | Temporal variegation govern AML relapse. (a) Clonal composition of 22 AMLs at diagnosis and relapse. Internally tangent circles represent successive lesions. Arrows indicate subclonal evolution inferred as indicated in the material and methods section. Delay from diagnosis to relapse is indicated in the upper right corner of each panel (m, months; y, years). (b) Numbers of stable and variegating lesions between diagnosis and relapse samples in the 22 AMLs. Colours are as in (a) with TP53 mutations included in the group of 'other lesions'. (c) Changes in clonal composition as a function of time to relapse. Droplets above and below the x axis represent the number of variegating and stable lesions, respectively. Paired droplets of single patients are connected by vertical lines. Droplets outlined by dashed lines indicate lesions from one patient whose relapse lesions (top droplet) were all different from diagnostic ones (bottom droplet). Grey circles indicate cases with no changes in clonal composition.

pre-leukaemic states, with evolutionary potentials similar to those of the ancestral clones.

According to these observations, one may expect early relapses developed from unchanged clones to be drug resistant, and late relapses emerging from the pre-leukaemic pool to be drug sensitive. We investigated this in the 19 out of 22 relapse AMLs who received intensive chemotherapy or 5-azacytidine as salvage therapy (Supplementary Table 1). We did not find any significant association between the response to the second treatment and either the clonal variation or the delay to relapse. Larger studies are mandatory to fully address this question.

To assess the persistence of relapse reservoirs after treatment, we performed Sanger sequencing (Fig. 3a), FISH (Fig. 3b), quantitative reverse transcription–PCR or deep sequencing analyses (Fig. 3c,d) in remission samples from 12 patients, including two (UPN2014-022 and UPN2014-041) who were still in remission 21 and 27 months after diagnosis, respectively, and 10 who further experienced relapse. Early *DNMT3A,* but also *TET2* and *ASXL1* mutations, as well as *MLL* and del(20q) rearrangements were all detected in these samples, whereas other lesions were inconstantly present (Fig. 3, Supplementary Table 8).

Taken together, these results show that cells carrying these early initiating lesions are frequently not eradicated by the treatment, and are able to re-initiate the disease.

**NSG repopulation by AML cells with early initiating lesions.** The pre-leukaemic potential of AML lesions can be assessed in xenotransplantation models[7,8]. We thus injected $5 \times 10^6$ T-depleted mononuclear cells from 38 AML blood samples, including 19 samples from our prospective cohort and 19 from other patients (Supplementary Fig. 1), to sublethally irradiated NSG mice to analyse their repopulation capacities with respect to cytogenetic and mutational patterns. Among these samples, 13 led to overt leukaemic engraftment with a clonal composition of human cell population remarkably mirroring the one of injected samples (Fig. 4a, Supplementary Figs 5 and 6, Supplementary Table 9). Fifteen samples repopulated NSG bone marrow with more than 0.5% of human CD45$^+$ (hCD45$^+$) cells but no overt leukaemia (Fig. 4a,b). Lymphoid or lympho-myeloid repopulation was detected in 13 of these 28 engrafting samples, including samples with *DNMT3A, TET2, ASXL1* mutations, CBF or *MLL* rearrangements at injection. In total, of the 38 AML samples, most samples with *TET2* mutations (7/8), *DNMT3A* mutations (9/12), *ASXL1* mutations (6 /7) and *MLL* fusions (3/4) led to either leukaemic or non-leukaemic repopulation (Fig. 4a,b, Supplementary Figs 5 and 6).

As *TET2* mutations were associated with *DNMT3A* or *ASXL1* mutations in 6/7 engrafting samples, and preceded *DNMT3A* mutations in some patients, we next asked whether isolated TET2 loss could induce pre-leukaemic expansion. CD34$^+$ cells from cord blood were transduced with lentiviruses expressing small hairpin RNA (shRNA) designed to knockdown TET2 or scramble control[23] and injected into NSG mice. Four months after injection, TET2 knocked-down cells showed a 2.6-fold greater repopulation (mean ± s.e.m.: 50 ± 6% of hCD45$^+$ cells in mouse bone marrow) than control cells (19 ± 5%), with both lymphoid and myeloid potentials (Fig. 4c,d, Supplementary Fig. 7a). Moreover, TET2 knocked-down human cells sorted from primary recipients' bone marrow led to lympho-myeloid repopulation 12 weeks after injection into secondary recipients (Supplementary Fig. 7b). This result shows that TET2 depletion is sufficient to improve the multi-lineage repopulation of NSG bone marrow with intact self-renewing capacity, a signature of pre-leukaemic stem cell function, as previously defined for *DNMT3A* mutations[8]. It has been proposed that the expansion of HSPCs with *DNMT3A* mutations favours subsequent accumulation of

additional mutations, driving progression to AML[8]. Altogether, our xenotransplantation results support this model for other putative pre-leukaemic lesions, including *TET2* mutations, *ASXL1* mutations, and *MLL* rearrangements.

**Clonal composition defines distinct AML genetic hierarchies.** By combining our data, we then attempted to reconstruct the clonal phylogeny of all our AML samples (Supplementary Fig. 8). When available, single-cell-derived colony analyses confirmed in most cases the order of lesions inferred from cytogenetic, molecular and sequencing data obtained on the bulk material (Supplementary Tables 10 and 11) indicating a strong correlation between the order of events inferred from variant allele frequencies (VAFs) and those inferred from colony analysis. In 49/72 cases at diagnosis, the first events observed were mutations in epigenetic regulators, CBF translocations, *MLL* rearrangements or chromosome 20q deletions, with a frequent ($n = 27$) early accumulation of epigenetic events in a dominant clone. This first hit was recurrently followed by lesion affecting *NPM1,* transcription factors, or the splicing machinery, and then mutations in signalling pathways (Fig. 5a, Supplementary Table 10).

Depending on initial lesions, distinct genetic hierarchies were delineated (Fig. 5b). First, 27/72 patients had a genetic hierarchy reminiscent of CHIP, with *DNMT3A, TET2* or *ASXL1* mutations co-occurring with either mutations in *NPM1* or in major hematopoietic transcription factors: *RUNX1, CEBPA,* and *GATA2.* In a second group of 10 patients, we found mutations in *NPM1* or in these transcription factors, but no mutations in *DNMT3A, TET2* or *ASXL1.* A third group of 14 patients had CBF, *MLL* or chromosome 20q rearrangements in first position, but no mutations in *NPM1* or in the aforementioned transcription factors. Finally, in 21/72 patients, neither CBF, *MLL,* 20q rearrangements nor mutations in *NPM1, RUNX1, GATA2* or *CEBPA* were retrieved. Beside cases with germline variants predisposing to AML—three *DDX41* mutations[24], one *CSF3R* mutation and one 14q32.2 duplication[25]—, this group comprised all seven patients with *TP53* mutations (Fig. 5b, Supplementary Fig. 9). Within the latter cases, we found co-occurring *DNMT3A* mutations in three out of four *de novo* AMLs but not in two therapy-related AMLs and one secondary AML. As VAFs indicated that *TP53* and *DNMT3A* mutations occurred in a unique dominant clone (Supplementary Tables 3 and 11), we analysed 155 and 51 single-cell-derived colonies from patients UPN2014-027 and UPN2015-027 (Fig. 6a,b). We failed in conclusively determining which of the *TP53* or *DNMT3A* mutations occurred first because all 138 and 51 mutant colonies from the two patients, respectively, had both mutations. These results contrasted with those previously obtained from the seven *DNMT3A* or *TET2* mutant AMLs with normal karyotype, where subsets of colonies with single founding mutations were detected, as a reflection of an expansion step preceding the acquisition of the second event (Fig. 6c).

**Discussion**
In the present study, we reconstructed the clonal composition of AML with a focus on known driver lesions and their putative pre-leukaemic initiating potential. We identify distinct routes for AML initiation and evolution in adults. The main one involves mutations in epigenetic regulators, such as *DNMT3A, TET2* and *ASXL1.* Lesions in these genes have been detected in 68% of individuals with CHIP and haematologic driver mutations[3–5] but in smaller proportions of *de novo* non-promyelocytic AMLs in our series (31/59, 53%) and in The Cancer Genome Atlas study[2] (66/180, 37%). This suggests that several other pre-leukaemic expanding lesions initiate other AMLs. First, del(20q) may be a

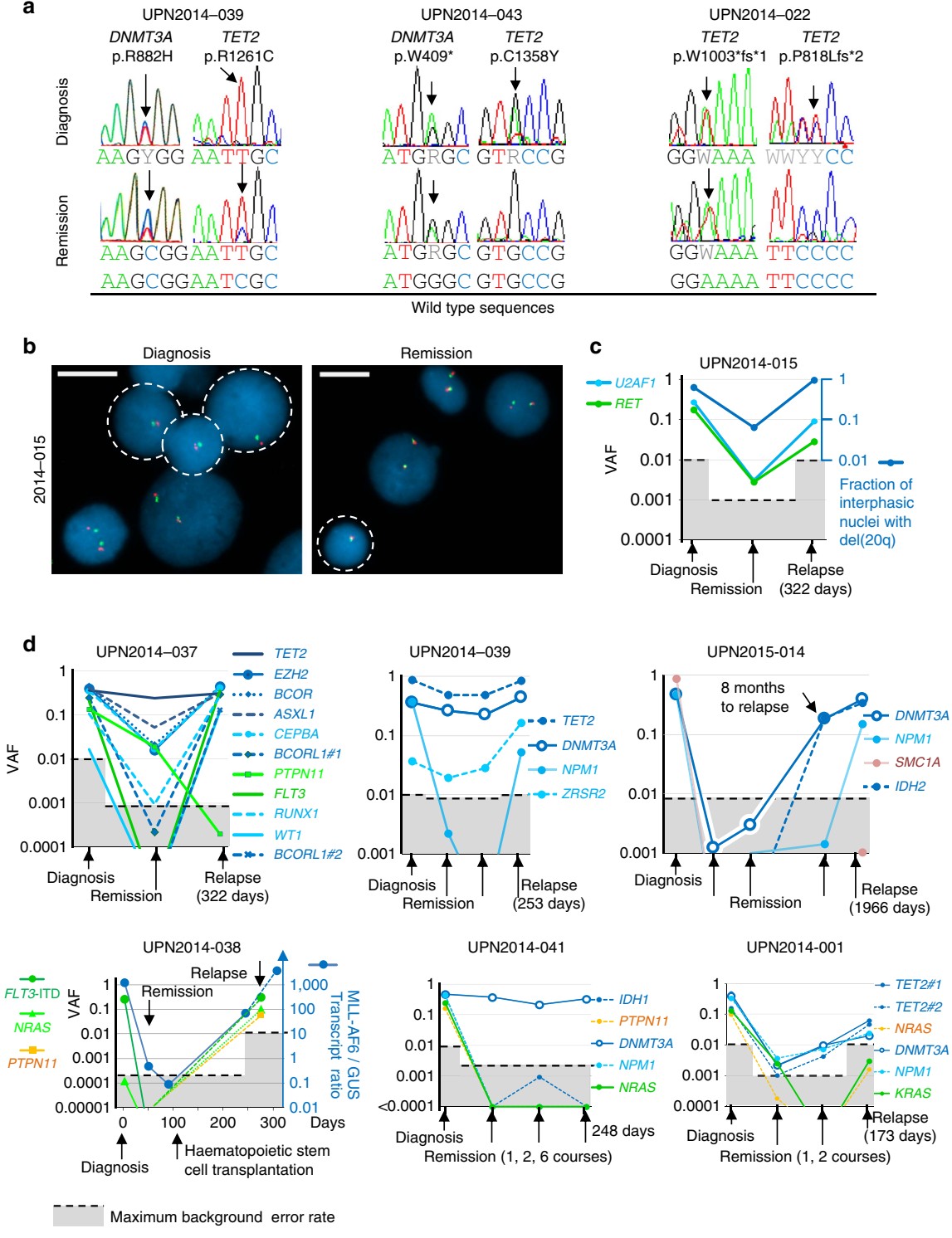

**Figure 3 | Retention of lesions in remission samples defines reservoirs for persistent clonal haematopoiesis and relapse** (**a**) Sequence traces of paired diagnostic and remission samples from three AML patients with *DNMT3A* and *TET2* mutations. Arrows indicate the detection of mutant alleles. (**b**) FISH analysis of diagnosis and remission samples from patient UPN2014-015 with persistence of del(20q). Presence of one instead of two red + green signals indicates del(20q)(outlined nuclei). Bars indicate 10 μm. (**c**) Kinetics of fraction of interphasic nuclei with del(20q) and VAFs of *U2AF1* and *RET* mutations determined by targeted or deep sequencing in samples from patient UPN2014-015. Black dashed lines over shaded areas delimit the maximum background error rate for single nucleotide variant detection of patient specific mutations by deep sequencing, and the 0.01 limit of detection for the gene panel targeted sequencing. Different time points of follow-up are marked by arrows. (**d**) Kinetics of lesions in six additional patients as in (**c**). MLL-AF6 transcript quantification in patient UPN2014-038 was monitored by quantitative reverse transcription–PCR with a limit of detection of 0.01.

*bona fide* CHIP lesion, as it was found in non-tumoral cells from patients with multiple myeloma devoid of myeloid malignancy[26]. Second, *RUNX1/RUNX1T1* translocations were detected in

Guthrie cards from new-borns who developed AML several years later[27]. Third, both CBF and *MLL* translocations promote non-leukaemic repopulation of immunocompromised mice by

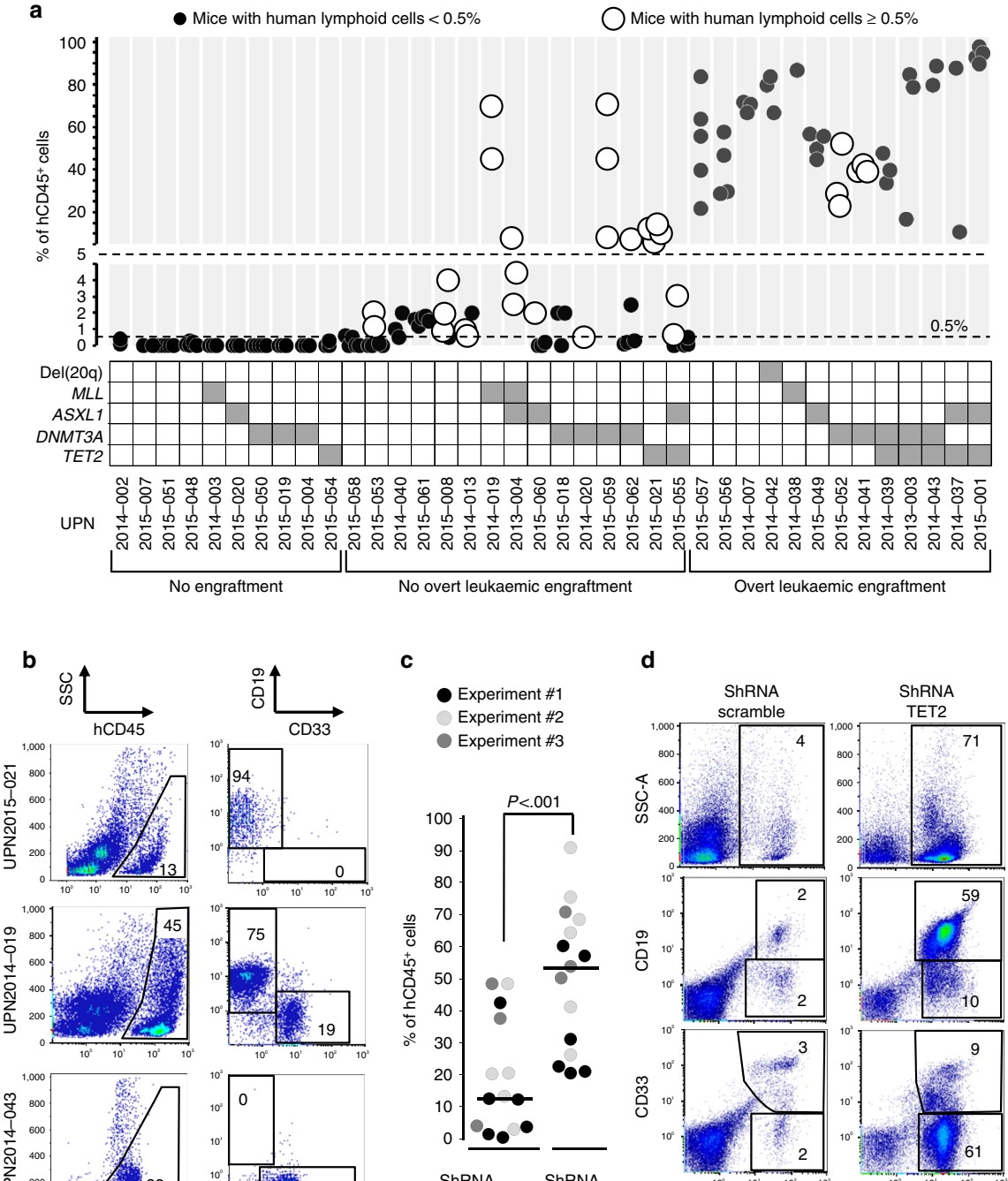

**Figure 4 | AMLs with early epigenetic lesions have leukaemic and non-leukaemic repopulation capacities in NSG mice. (a)** Percentages of human CD45[+] (hCD45[+]) cells in the bone marrow of NSG mice 8–43 weeks after injection of $5 \times 10^6$ mononuclear cells from 38 AML blood samples. The presence of candidate pre-leukaemic lesions in injected cells is indicated in the co-mutation table. **(b)** Flow cytometric analysis of NSG bone marrow repopulated with non-leukaemic (UPN2014-019, UPN2015-021) and leukaemic (UPN2014-043) cells. **(c)** Repopulation of NSG bone marrow by shRNA TET2 or scramble transduced cord blood CD34[+] cells. Bars indicate the median (Mann–Whitney test). **(d)** Flow cytometric analysis of the bone marrow from two representative mice transplanted with control (shRNA scramble) and TET2 knocked-down (shRNA TET2) cells.

human HSPCs[12–14]. In line with these observations, we found that del(20q), CBF and *MLL* translocations are early AML events retained in the clone when patients experience relapse. Finally, in some AMLs without these lesions, we found early mutations in *IDH2*, which have also been proposed to be pre-leukaemic lesions[8,11], or early mutations in *SF3B1*, *U2AF1*, *TP53* and *JAK2*, all detected in a few individuals with CHIP[3,4]. In our work, we did not perform whole-genome sequencing but focused on 122

genes recurrently mutated in AML with additional whole exome sequencing in seven patients. Thus, we cannot exclude that some lesions missed by our strategy can be first drivers. However, our results recapitulate those from larger studies with whole-genome and whole-exome analyses[2]. Moreover, when analyzing non-promyelocytic AMLs from the TCGA cohort, we were able to retrieve the four genetic hierarchies found in our study with roughly similar proportions of cases.

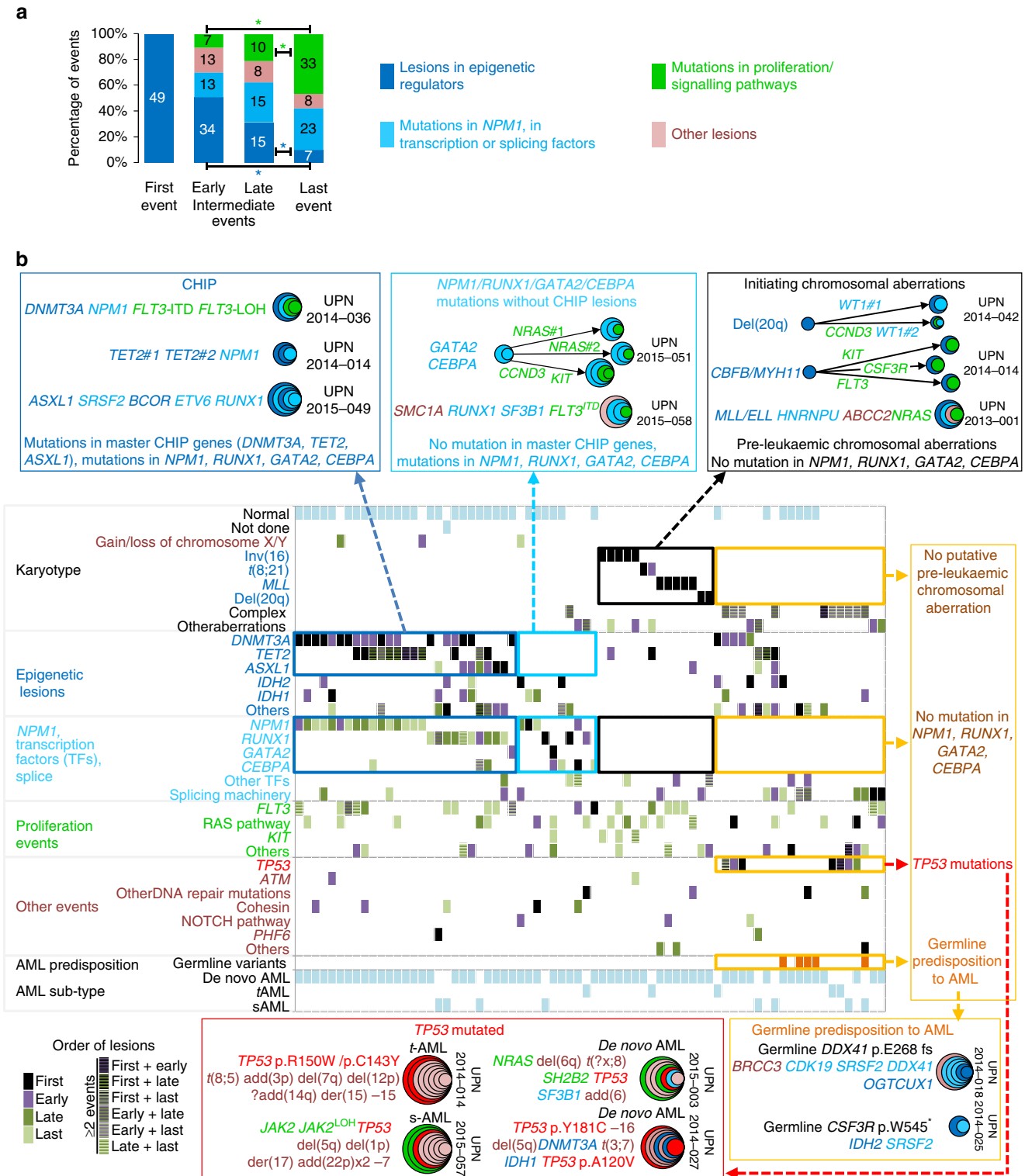

**Figure 5 | Founding lesions define distinct genetic hierarchies and clonal histories in AML. (a)** In 49 patients with lesions in epigenetic regulators first, subsequent events were classified as early intermediate (second to median), late intermediate (median +1 to penultimate) and last event. Histograms show the distribution of events at each chronological position. * indicate $P < 0.05$, Fisher's exact test. (**b**) Co-mutation table of 72 AMLs at diagnosis. The bottom left colour code indicates the position of each lesion as defined in **a**; hatched boxes mean ≥2 lesions at distinct positions. Groups of patients with distinct genetic hierarchies were defined according to the co-occurrence or exclusion of mutations in the three master genes involved in CHIP—*DNMT3A*, *TET2*, *ASXL1*—, mutations in *NPM1* and in haematopoietic transcription factors—*RUNX1*, *GATA2*, *CEBPA*— and pre-leukaemic chromosomal aberrations—*MLL* and CBF rearrangements, del(20q). Clonal composition of representative AMLs with distinct genetic hierarchies are shown in boxes surrounding the mutation table. s-AML, secondary AML; t-AML, therapy-related AML. Internally tangent circles represent successive events. Colours are as in Fig. 2a.

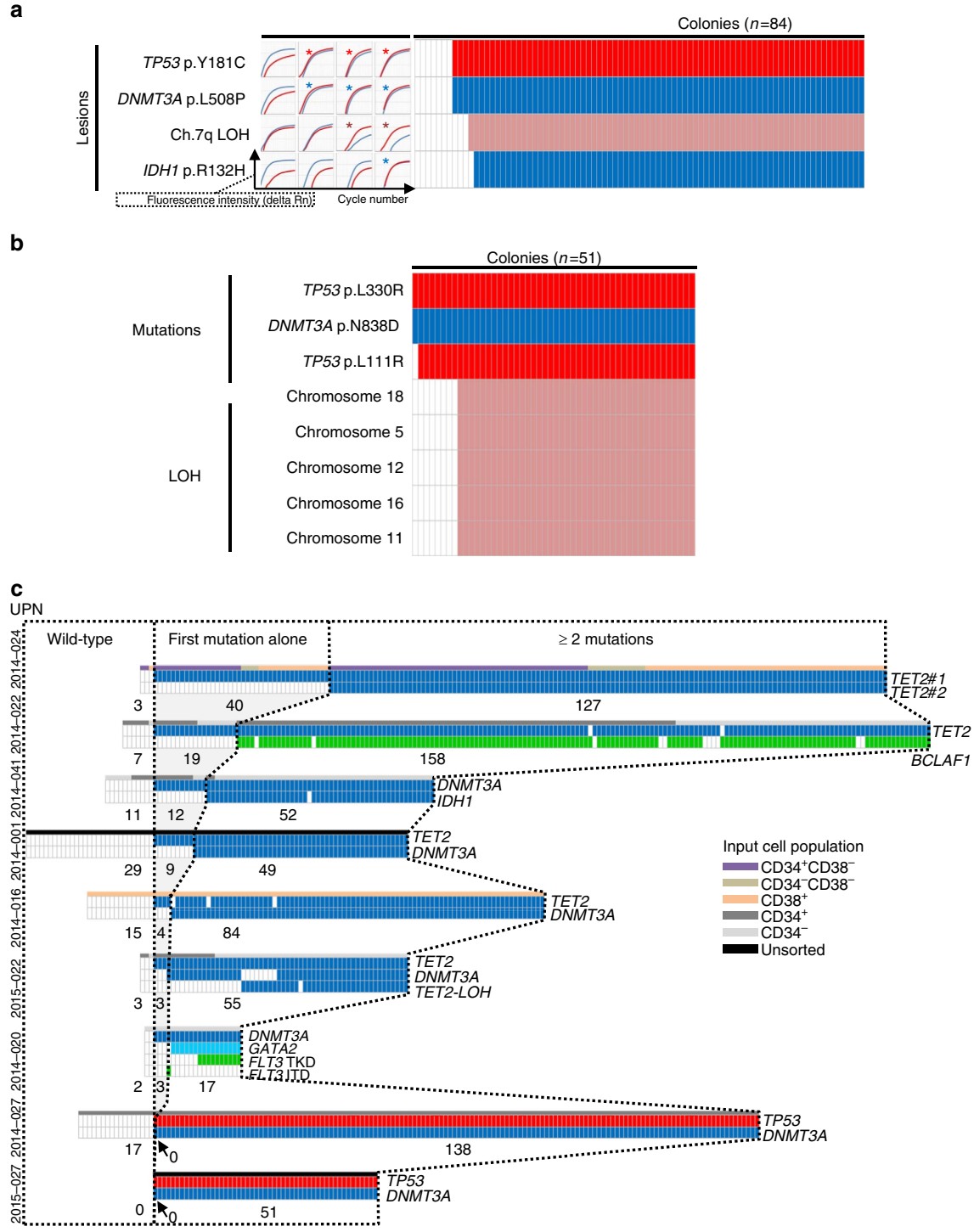

**Figure 6 | Capture of early clonal expansion steps by single-cell-derived colony analysis. (a,b)** Genotyping of single cell derived colonies from two *de novo* AMLs—UPN2014-027, (**a**) and UPN2015-027, (**b**)—with concomitant *DNMT3A* and *TP53* mutations, as in Fig. 1. Losses of heterozygosity (LOH) were detected using informative SNPs. Asterisks indicate mutant allele detection or LOH. (**c**) Mutational patterns of individual colonies from patients with early initiating lesions. Cell sorting strategies before culture are indicated above each table. Dotted lines separate wild-type colonies, colonies with one mutation, and colonies with ≥ 2 mutations/lesions. The number of colonies of each category is indicated below the tables.

Most of the aforementioned early lesions may lead to HSPC expansion and pre-leukaemic clonal haematopoiesis that requires additional hits for the onset of AML. In abnormally expanding HSPCs, the excess in mitoses may increase the absolute number of unrepaired replication errors, thereby favouring the accumulation of subsequent mutations. Depending on the genetic hierarchy of AML—that is, the nature of the initial lesions

and the number of additional mutations necessary to trigger a full blown disease—, distinct latencies from the onset of clonal haematopoiesis to AML may be expected. For instance, the typical CHIP-derived hierarchy of AML comprises early lesions in *DNMT3A*, *TET2* and *ASXL1* that aggregate together or with other mutations in epigenetic regulators, followed by mutations in *NPM1* or in haematopoietic transcription factors, and then by

lesions in signalling pathways such as FLT3 and RAS. In other AMLs, CBF and *MLL* rearrangements induce both epigenetic and haematopoietic transcription factor deregulation[15–17]. This may explain the requirement for less additional lesions, and may result in a shorter latency than in CHIP-derived diseases for the progression to acute leukaemia (Fig. 7a). Nevertheless, functional studies remain necessary to understand the mechanisms of lesion cooperation and their precise consequences in the dynamics of AML clones.

In one third of our patients, neither recurrent CBF or *MLL* translocations, nor typical CHIP hierarchy or mutations in *NPM1*, *RUNX1*, *CEBPA* and *GATA2* were found. This group of patients was enriched in cases with germline predisposition to AML and *TP53* mutated cases, including three *de novo* AMLs with concomitant *DNMT3A* mutations. *TP53* mutations have a major role in therapy-related AML, in which the previous exposure to chemotherapy for a first cancer leads to the preferential expansion of pre-existing heterozygous mutant HSPCs[28]. Such an exposure does not exist in *de novo* AMLs, and our analysis of *DNMT3A* and *TP53* mutations in single-cell-derived colonies suggests that co-occurring lesions may be essential to provide an expansion capacity to *TP53* mutant HSPCs. In line with this, among the 39 *TP53* mutations detected in 1,125 individuals with CHIP[3–5], nine had co-occurred with other mutations, including two in *DNMT3A* and four in *TET2*.

Finally, we established a link between the number of clonal changes and the delay from diagnosis to relapse. In our AML series, as in other series of AMLs and acute lymphoblastic leukaemias[29,30], relapses arose from the persistence of a clone in which a few lesions changed, or in which only the first founding one was retained. In this last situation, we observed that some ancestral clones were able to re-evolve in a way similar to the one that built the initial genetic hierarchy, triggering relapse up to 10 years after the first AML diagnosis. In patients who achieve complete remission (CR), this return to a pre-leukaemic condition raises the question whether the clone will progressively

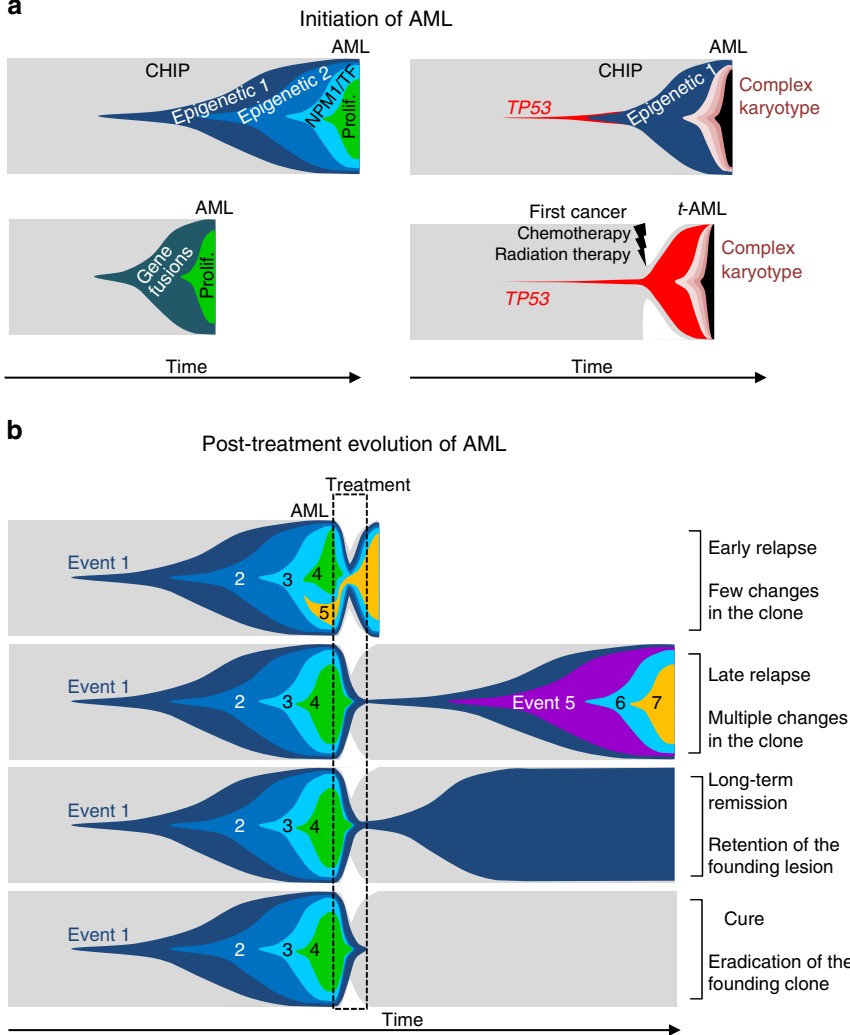

**Figure 7 | The action of time and treatment selection pressure in the initiation and evolution of AML. (a)** Schematic representation of the initiation and evolution of the malignant clone (coloured areas) at the expense of normal HSPCs (grey area) from the acquisition of the first event to the onset of AML. Left diagrams show AMLs emerging after the occurrence of the first lesions which endow HSPCs with enhanced expansion capacity. Right diagrams depict AMLs with *TP53* mutations. For therapy-related AMLs (t-AML), the occurrence and treatment of a first cancer, followed by HSPC depletion and recovery, are indicated. The nature of each lesion is indicated as follows: Epigenetic 1: mutations in *DNMT3A*, *TET2*, *ASXL1*; Epigenetic 2: Epigenetic 1 or other mutations in epigenetic modifiers; Gene fusions: CBF or *MLL* translocations; *NPM1*/TF: mutations in *NPM1* or haematopoietic transcription factors; Prolif.: mutation in proliferation/signalling pathways; *TP53*: *TP53* mutations. **(b)** Post-treatment evolution of AML, according to various outcomes. CHIP, clonal haematopoiesis of indeterminate potential.

re-accumulate new driver lesions or will stay dormant as seen in the vast majority of individuals with CHIP who never develop any haematopoietic malignancy (Fig. 7b). Understanding the mechanisms by which the clone will leave dormancy to re-initiate leukaemia may provide powerful biomarkers and therapeutic targets to detect and prevent late relapses.

## Methods

**Samples.** Bone marrow (BM) and blood samples were collected at the time of diagnosis, CR and relapse from AML patients after informed consent in accordance with the declaration of Helsinki and the local ethics committee of Saint-Antoine hospital. Mononuclear cells were obtained after ficoll separation and cryopreserved at the tumour bank of Saint-Antoine Hospital, Paris. Fresh or frozen samples were used for further cellular and molecular experiments. BM cells were sorted to enrich the samples in leukaemic or non-leukaemic progenitor cells. $CD34^+$, $CD38^+$, $CD34^-$ or $CD34^+CD38^-$ cell sorting strategies were established according to initial leukaemia immuno-phenotype. Blood cells from CR were sorted to isolate the $CD3^+$ fraction. Immuno-magnetic cell sorting was performed using CD3, CD34 or CD38 MicroBeads and columns (MACS, Miltenyi Biotec) according to the manufacturer's protocols.

Umbilical cord blood samples were collected from healthy new-borns with mothers' consent in accordance with the institutional review boards of the Etablissement Français du Sang, Créteil France, and the Institut National de la Santé et de la Recherche Médicale, Paris, France. $CD34^+$ cells were isolated using a dextran/ficoll based procedure followed by immuno-magnetic separation (MACS, Miltenyi Biotec).

**Targeted sequencing.** A panel of 122 genes mutated in AML and myeloid malignancies was designed (Supplementary Table 2). Amplicon libraries were obtained from 225 ng of BM DNA, using HaloPlex Target Enrichment System (Agilent technologies), according to the manufacturer's protocol. Sequencing was performed using a MiSeq sequencer (Illumina) using the manufacturer recommendations. Raw data from targeted sequencing have been deposited at EGA and are available on request (EGA study number: EGAS00001001779).

Results were analysed after alignment of the reads using the SureCall Software version 3.0.1.4 from Agilent Technologies. BWA MEM algorithm was used for alignment and Illumina SNPPET SNP Caller algorithm was used to identify single nucleotide polymorphism (SNP) and indel variants. Minimum allele frequency for variant calling was set at 5% with a minimum local depth at 40. All variants were manually checked using IGV 2.3 software. Variants identified in diagnosis or relapse samples were manually checked in paired relapse or diagnosis samples using IGV software to study clonal evolution between samples and to rule out low frequency variants. For all samples, average depth in target regions was 619 (range 270–1650) and 98.26% (range 93.5–99.5) of target regions were covered by at least 100 reads.

Detection of described polymorphism was performed by an in-house software using Ensembl database. Non-described variants of potential interest with VAF >10% were sequenced using the Sanger method in the diagnosis BM sample and in the $CD3^+$ fraction from CR or non-sorted CR samples, in order to rule out artifacts and non-somatic variants. All primers are described in Supplementary Information (Supplementary Table 12).

**Exome sequencing.** Exome sequencing was done after sequence capture, enrichment and elution according to the manufacturer's instructions (SureSelect, Agilent) without modification except for library preparation (NEBNext Ultra kit, New England Biolabs): 600 ng of each genomic DNA were fragmented by sonication and purified to yield fragments of 150–200 bp. Paired-end adaptor oligonucleotides from the NEBNext Ultra kit were ligated on repaired A tailed fragments, then purified and enriched by eight PCR cycles. In all, 1,200 ng of these purified libraries were then hybridized to the SureSelect oligo probe capture library for 72 h. After hybridization, washing and elution, the eluted fraction was PCR-amplified with nine cycles, purified and quantified by QPCR to obtain sufficient DNA template for downstream applications. Each eluted-enriched DNA sample was then sequenced on an Illumina HISEQ 2000 as paired-end 75b reads. Image analysis and base calling was performed using Illumina Real Time Analysis (RTA 1.17.21.3) with default parameters.

The bioinfomatic analysis of sequencing data was based on the Illumina pipeline (CASAVA 1.8.2). CASAVA performs alignment of a sequencing run to a reference genome (hg19), calls the SNPs based on the allele calls and read depth, and detects variants (SNPs and Indels). The alignment algorithm used is ELANDv2 (performs multiseed and gapped alignments). Only the positions included in the bait coordinates were conserved. Genetic variation annotation was realized from IntegraGen in-house pipeline, consisting in annotation of genes (RefSeq) and known polymorphisms (dbSNP 132, 1000Genome, EVS) followed by a mutation characterization (exonic, intronic, silent, nonsense….). For each position, the exomic frequencies (homozygous and heterozygous) were determined from all the exomes already sequenced at Integragen, and the exome results provided by 1000G, EVS and HapMap. Minimum average depth on the exome is around $\times 70$. Samples

from diagnosis and CR were paired-analysed and manually checked with IGV v2.3 to identify somatic variants. Confirmed somatic variants are shown in supplemental data. Raw data from exome sequencing have been deposited at EGA and are available on request (EGA study number: EGAS00001001779).

**Methyl cellulose assays.** Sorted cells from diagnostic samples were seeded in triplicate at 100 to 300,000 cells per 1 ml culture dish in 2% standard methylcellulose medium (Stem Cells Technologies) supplemented with 37% fetal calf serum, 12% bovine serum albumin, 1% L-glutamine, $10^{-6}$ M β-mercaptoethanol, $1 IU ml^{-1}$ of erythropoietin, $50 ng ml^{-1}$ of stem cell factor, $25 ng ml^{-1}$ of FLT3 ligand, $10 ng ml^{-1}$ of interleukin 3, $10 ng ml^{-1}$ of interleukin 6, $10 ng ml^{-1}$ of granulocyte-colony stimulating factor, $5 ng ml^{-1}$ of granulocyte-macrophage colony-stimulating factor, $10 ng ml^{-1}$ of thrombopoietin, $100 IU ml^{-1}$ penicillin and $100 µg ml^{-1}$ streptomycin. Colonies were counted after 14 days and picked. When appropriate, after a wash in PBS, individual colonies were splitted into two cell suspensions to perform both genotyping and FISH analyses. The first cell suspension was gently deposited onto 18-well immunofluorescence slides. Slides were dried and fixed for further FISH analyses. The second cell suspension, or whole colonies when no translocation was detected in bulk AML, was stored at $-80 °C$ for further geno-typing assays.

**Single-cell-derived colony genotyping assays.** DNA from individual colonies was prepared from a 50 µl lysis buffer containing 0.5 µl tween 20, 18.5 µg recombinant proteinase K (Thermo scientific) and $H_2O$, after incubation at $56 °C$ for 1 h and $95 °C$ for 15 min. Genotyping was performed using custom Taqman SNP genotyping assays (Life technologies) with 5 µl colony DNA, using a 7,500 fast real-time PCR system (Applied Biosystem), according to the manufacturer's protocol. For chromosomal deletion investigation, SNP genotyping assays of described SNP of the deleted area were performed, using MYBL1 p.I624M for del(20q), FANCA p.G809D for del(16q) and CUX1 p.A418T for del(7). Sequences of probes and primers are described in Supplementary Information (Supplementary Table 13).

**FLT3-ITD and HNRNPU large indel detection.** The mutational status for FLT3-ITD and HNRNPU at diagnosis and in colonies was determined using high-resolution sizing of fluorescent dye-labelled PCR amplification. Beyond its contribution to mutation detection, high-resolution sizing also allowed to estimate the allelic burden by measuring peak height ratios (mutant/wild-type + mutant) on a fluorescence scale. PCR probes and primers are described in Supplementary Information (Supplementary Table 12).

**Cytogenetic and FISH analyses.** Conventional cytogenetic analysis was performed in diagnosis and relapse samples on R-banding metaphases obtained from 24 h unstimulated culture using standard procedures. Karyotypes were interpreted according to the recommendations of the International System for Human Cytogenetic Nomenclature. Chromosomal rearrangements were confirmed by FISH, using the following probes: MLL breakapart probe (LPH 013), AML1 (RUNX1) breakapart probe (LPH 027), del (5q) probe (LPH 024), del(7q) deletion probe (LPH 025), del(20q) deletion probe (LPH 020), inv(16) probe (LPH 022), fast FISH X, Y an 18 (LPF 002) (all from Cytocell Ltd, Cambridge UK). NUP98 (11p15) break probe (Kreatech), Vysis LSI D7S522 (7q31)/CEP7 dual colour probe (Abbott) and XL 7q22 / 7q36 deletion (MetaSystems) were also used. Hybridization signals were scored on a BX61 fluorescence microscope with an UPLAN FLN $100 \times /1.30$ oil immersion lens (Olympus, Rungis, France) in 176–500 nuclei and analysed using CytoLabView BandView/FISHView Analysis (VDS)—6.0 software (Applied Spectral Imaging Ltd, Yokneam, Israel). The same FISH analyses were performed on colonies, except that fewer nuclei were analysed.

**MLL/AF6 transcript detection and quantification.** Total RNA was isolated from Ficoll isolated mononuclear cells by use of isothiocyanate guanidium method (Extract All, Eurobio, France). RNA (1 ug) was reverse transcribed using Moloney murine leukaemia virus reverse transcriptase. complementary DNA synthesis was performed with the following primers (forward: 5′-GAGGATCCTGCCCCAAAG AAAAG-3′; reverse: 5′-GGGAGAGGACAGCATTCGC-3′). Sanger sequencing of the PCR product was performed in order to confirm the MLL and AF6 exons implicated into the fusion transcript. The patient follow-up was monitored using real time quantitative PCR of the MLL-AF6 fusion according to the European Leuke-mianet recommendation[31], using the following primers and probe (forward: 5′-GTCCAGAGCAGAGCAAACAGAAA-3′; reverse: 5′-GAAAATAAAATCTCA TCACTCCATGG-3′; probe: 5′-CTCCCCGCCCAAGTATCCCTGTAAA-3′) adapted from published data[32] and normalized to GUS reference gene expression. Sensitivity reached $10^{-5}$.

**CBFB/MYH11 transcript detection and quantification.** CBFB/MYH11 transcript detection and quantification were performed according to the European Leukemianet recommendation[31].

**Deep targeted sequencing.** A targeted-resequencing panel including all variants detected at diagnosis and relapse in 11 patients was designed. Amplicon libraries were obtained from 57.6 ng of DNA, using HaloPlex HS Target Enrichment System (Agilent technologies), according to the manufacturer protocol. With this method, after an initial enzymatic digestion, DNA fragments are captured with target probes, and hybridized with two indexes. The first index is sample specific. The second index is a unique random sequence of 10 nucleotides, with a specific sequence for each DNA fragment of the sample. After PCR amplification at the end of library preparation, amplicon families bearing the same indexes can be identified, allowing a better detection of PCR and sequencing errors, and a better sensitivity than standard HaloPlex target enrichment system. Sequencing was performed using a MiSeq sequencer (Illumina) using manufacturer recommendations. Results were analysed after alignment of the reads using the SureCall Software version 3.0.2.1 from Agilent Technologies. Variants detected at diagnosis and relapse were manually checked in the CR sample, using IGV software version 2.3 in order to identify low frequency variants. The background error rate at a single nucleotide variant position was calculated as the ratio of the sum of amplicon families with 'non-reference' or 'non-mutant' bases to the total number of amplicon families at this position. For all patient-specific samples, the maximum background error rate was defined as the maximum value of all background error rates obtained at target positions. As indels are not subject to sequencing error-dependent miscalling, the sensitivity of indel detection was set as 1/(number of amplicon families).

**Xenograft experiments with AML cells.** NSG mice were bred and maintained under specific pathogen free conditions with acidified water (pH 5.3) at the animal facility of Gustave Roussy Institute. Animal experiments were performed in accordance with guidelines established by the Institutional Animal Committee. Peripheral blood mononuclear cells from AML patients were depleted in $CD3^+$ cells by RosetteSep human $CD3^+$ depletion cocktail (StemCell Technologies) and $5 \times 10^6$ cells were intravenously injected to female mice (6–8 weeks old) 24 h after irradiation at 2.5 Gy from a $^{137}Cs$ source[33]. Mice were analysed at 8–43 weeks post-injection. Cells from mouse BM were stained with rat anti-mouse CD45 (Biolegend) and mouse anti-human CD45, anti-human CD19, anti-human CD33 and anti-human CD3 antibodies (all from BD Pharmingen; clones and fluorochromes are indicated in Supplementary Table 12). Stained cells were analysed on FACSort or FACSCanto II cytometers (BD Biosciences). The presence of <0.5% of human $CD45^+$ ($hCD45^+$) population was considered as non-engraftment. The presence of >5% of $hCD45^+$ cells with major $CD33^+$ population (>75% of the $hCD45^+$ cells) was considered as overt AML engraftment. The presence of 0.5 to 5% of $hCD45^+$ or the presence of >5% of $hCD45^+$ with <75% of $CD33^+$ in the $hCD45^+$ population was considered as non-overt or non-leukaemic engraftment.

**Xenograft experiments with TET2-depleted $CD34^+$ cells.** $CD34^+$ cells from three to ten distinct cord blood samples were pooled and transduced with lentiviruses (pRRLsin-PGK-eGFP-WPRE, Genethon, Evry, France) expressing the green fluorescent protein and either a short hairpin RNA targeting TET2 (shRNA-TET2, 5′-GGGTAAGCCAAGAAAGAAA-3′) or a scramble sequence (shRNA-scramble, 5′-GCCGGCAGCTAGCGACGCCAT-3′) as control[23]. Twenty-four hours after transduction, $2 \times 10^5$ cells were intravenously injected to sublethally irradiated NSG female mice (6–8 weeks old). Mice were killed 15–17 weeks after injection, and repopulation of mouse bone marrow (femurs and tibias) by human cells was assessed by flow cytometry, using APC-conjugated mouse anti-human CD45, PE-conjugated anti-human CD19, PE-conjugated anti-CD33 (all from BD Pharmingen). Antibodies are listed in the Supplementary Information (Supplementary Table 14).

For NGS experiments and for secondary transplantation, bone marrow of repopulated mice was enriched in human cells using a mouse/human chimera isolation kit (Stem Cell Technologies).

**Determination of clonal composition.** To determine the order of chromosomal and genomic lesions in each AML diagnosis or relapse sample, we combined the following quantitative results: (1) frequencies of karyotype metaphases harbouring each chromosomal aberration, (2) frequencies of interphasic nuclei with specific translocations, rearrangements, deletions or gains, (3) VAFs from exome or targeted sequencing runs for somatic gene mutations (single nucleotide variants and indels), (4) VAFs from exome or targeted sequencing for SNPs in sequenced regions with LOHs or copy number variations, (5) Peak height ratios of high-resolution sizing of fluorescent dye-labelled PCR amplification for FLT3-ITDs. Except for FLT3-ITD peak height ratios, 95% confidence intervals were calculated by using the numbers of analysed nuclei and read depths as sample sizes for cytogenetic and NGS quantifications, respectively. These quantitative values were converted to fractions of cells harbouring the lesions (Variant Cell Fraction, VCF), taking into account LOHs, with or without copy number variations, as well as the gender of the patient for lesions on chromosome X. Then, lesions were ordered according to VCFs with correction in case of variant alleles involving genes mapped on chromosomes with imbalances or LOH. When available, we integrated the results from colony analyses performed at diagnosis and the order of events

from quantitative data from distinct time point samples (diagnosis/CR/relapse) to correct or refine the clonal phylogeny.

**Data availability.** Raw exome and targeted sequencing data have been deposited at the European Genome-phenome Archive (EGA, https://ega-archive.org) under accession number EGAS00001001779. All other relevant data is contained within the article or supplementary files, or available from the author upon request.

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

## Acknowledgements

This project was funded by the ARC foundation (N°EML20110602421), the Région Ile-de-France (N°2012-2-eml-06-UPMC_12016710), the Association Laurette Fugain (N°J15I409 to FD)(N°ALF/06-05 to F.L.), the Institut National du Cancer to F.L. (N°2010-1-RT-04) and to P.H., and the Institut Universitaire d'Ingénierie en Santé. We would like to thank The Cancer Genome Atlas (TCGA) for providing free access to their somatic mutational data. We would like to thank Simona Lapusan, Françoise Isnard, Anne-Claire Mamez, Paul Coppo for their help in collecting samples, Christelle Mazurier and Christine Nguyen for their technical advices and assistance, Mélanie Letexier, Jean-Paul Saraiva and Emmanuel Martin for exome sequencing and analysis, Nicole Casadevall and William Vainchenker for helpful discussions and critical reading of the manuscript.

## Author contributions

P.H. performed cell culture, genotyping and NGS experiments, interpreted the results and wrote the manuscript. Y.Z., F.L. performed xenograft experiments, interpreted the results and wrote the manuscript. V.J. performed xenograft experiments. R.T. designed NGS assays and participated in material collection. H.B., H. Moatti and P.F. contributed in cell culture experiments and genotyping of colonies. E.P., H.M. and A.B. performed shTET2 experiments. C.M. and D.B. performed standard molecular analysis. F.F. and F.L. contributed in material collection, sample preparation and in standard molecular analyses. R.F. performed *RUNX1* Sanger sequencing. F.F. performed flow cytometry analysis of patient samples. M.M. and O.L. contributed in follow-up of patients. C.B.N. performed cytogenetic and FISH analyses. F.D. designed the research, interpreted the results and wrote the manuscript. All authors contributed in manuscript review

## Additional information

**Competing financial interests:** The authors declare no conflict of interest.

