## [Peer Review File · Nature Communications]

Reviewer #1 (Remarks to the Author):

The authors have addressed all my comments, and in my view have also satisfactorily addressed the comments by the other two reviewers. I would recommend publication in Nature Communications.

Reviewer #4 (Remarks to the Author):

The manuscript by Hirsch and Zhang, et al. addressed the identification of pre-leukemic clone and genetic hierarchy of several AML subtypes using comprehensive mutational analysis and single-cell derived colony experiments. Although the overall topic is of interest and technically well done, there are several reports elucidating that DNMT3A/TET2/ASXL1 mutation, AML1/ETO, and MLL fusions are initiation events in AML as referred by the authors. It is an exaggeration that del(20q) is the first driver event because only one patient with del(20q) was included.

Reviewer #5 (Remarks to the Author):

The revised manuscript presents an analysis of 53 non-M3 AML cases, including inferred order of mutation acquisition at diagnosis, change in clonal composition at relapse, pattern of mutation clearance at remission, and in vivo repopulating activity. Comments:

1. It is difficult to track which samples were subjected to each of the many analyses in the paper (e.g., targeted vs. exome sequencing, colony genotyping, serial analysis at relapse, xenotransplantation, paired diagnosis/remission/relapse). A simple flowchart as a supplementary figures would make it easier to navigate the paper.
2. There is some inconsistency in the way pre-leukemic stem cells are defined in the paper. In particular, the part of the operational definition related to in vivo repopulating activity in different sections of the manuscript implies that 1) they should contribute to initiation of leukemia in vivo, 2) that they should confer multilineage potential in vivo, or 3) that they should be present in cells that retain multilineage potential. My impression is that there is more consensus around the latter version, although this is controversial. In any case, a single consistent operational definition should be used throughout.
4. Consider adopting the term "CHIP" (clonal hematopoiesis of indeterminate potential) which has gained more traction in the field than "ARCH."
5. The following statement: "...larger studies with whole genome and whole exome analyses where additional variants were mostly considered as passenger mutations." is overstated. While it is true that the greater number of variants detected by exome and/or genome sequencing in other studies have been generally used to define clonal architecture with greater precision, most publications are careful to point out that biological significance (ie, passenger vs. driver) requires either functional analysis or a demonstration of recurrence (or both). The sentence is sufficient with the clause "...where additional variants..." omitted.

Response to reviewers

We thank reviewers #1, #4, and #5 for their useful comment. Our point-by-point response appears in blue in the text.

REVIEWERS' COMMENTS:

Reviewer #1 (Remarks to the Author):

The authors have addressed all my comments, and in my view have also satisfactorily addressed the comments by the other two reviewers. I would recommend publication in Nature Communications.

We thank reviewer #1 for his/her comments that allowed us to substantially improve our manuscript.

Reviewer #4 (Remarks to the Author):

The manuscript by Hirsch and Zhang, et al. addressed the identification of pre-leukemic clone and genetic hierarchy of several AML subtypes using comprehensive mutational analysis and single-cell derived colony experiments. Although the overall topic is of interest and technically well done, there are several reports elucidating that DNMT3A/TET2/ASXL1 mutation, AML1/ETO, and MLL fusions are initiation events in AML as referred by the authors. It is an exaggeration that del(20q) is the first driver event because only one patient with del(20q) was included.

As already stated in our previous response to referees #2 and #3, the aim of our work was to generalize the concept of initiating lesions and subsequent genetic hierarchies to a series of AML reflecting the genetic variety of this disease. We did that by analyzing the order of events, the retention of early lesions in relapse and remission samples, and by performing in vivo reference assay (xenotransplantation to NSG mice). We agree this is not new for individual lesions, but we feel that our contribution helps generalizing the concept, and besides, this allows us to pinpoint particular genetic hierarchies enriched in cases with *TP53* mutations or AML predisposition (Fig.5b). Regarding del(20q), to our knowledge, very few studies have addressed the question of the order of events in myeloid malignancies with isolated del(20q). For instance, del(20q) was found to occur either before or after JAK2V617F mutation in myeloproliferative neoplasms (Kralovics et al, Blood, 2006; Schaub et al, Blood, 2009; Beer et al, Blood, 2010), and serial analyses of five myelodysplastic syndromes with co-occurring *U2AF1* mutations suggested in 1/5 cases that del(20q) was the second event (Bacher et al, BJH, 2014). In our AML study, we did not analyze only one patient but two patients (UPN14-015 and UPN14-042), in whom we formally demonstrate that del(20q) is the first event by colony assays in one patient, and by serial sample analysis in the second patient. We agree the number of cases is small, but del(20q) was shown to be present in non-tumoral cells from the bone marrow of patients with multiple myeloma, indicating it may behave as a pre-leukaemic CHIP lesion as stated in the discussion.

Reviewer #5 (Remarks to the Author):

The revised manuscript presents an analysis of 53 non-M3 AML cases, including inferred order of mutation acquisition at diagnosis, change in clonal composition at relapse, pattern of mutation clearance at remission, and in vivo repopulating activity. Comments:

1. It is difficult to track which samples were subjected to each of the many analyses in the paper (e.g., targeted vs. exome sequencing, colony genotyping, serial analysis at relapse,

xenotransplantation, paired diagnosis/remission/relapse). A simple flowchart as a supplementary figure would make it easier to navigate the paper.

We added a flowchart in the supplementary information (Supplementary Fig. 1). The numbers of prospectively and retrospectively included patients were re-assessed and corrected in accordance.

2. There is some inconsistency in the way pre-leukemic stem cells are defined in the paper. In particular, the part of the operational definition related to in vivo repopulating activity in different sections of the manuscript implies that 1) they should contribute to initiation of leukemia in vivo, 2) that they should confer multilineage potential in vivo, or 3) that they should be present in cells that retain multilineage potential. My impression is that there is more consensus around the latter version, although this is controversial. In any case, a single consistent operational definition should be used throughout.

We tried to use a more homogeneous definition of pre-leukaemic cells throughout the manuscript by adding the "multilineage" notion where it was initially missing:

- In the abstract :

*"In acute myeloid leukaemia (AML) initiating pre-leukaemic lesions can be identified through three major hallmarks: their early occurrence in the clone, their persistence at relapse, and their ability to initiate **multilineage haematopoietic repopulation** and leukaemia in vivo".*

- In the introduction :

*"three major hallmarks of pre-leukaemic initiating events : 1) their early occurrence in the clone 7,8,10,11 , 2) their persistence at relapse 8,11, and 3) their ability to provide a **multilineage** selective advantage to mutant over normal HSPCs in vivo 7,8."*

4. Consider adopting the term "CHIP" (clonal hematopoiesis of indeterminate potential) which has gained more traction in the field than "ARCH."

We changed ARCH to CHIP in the text and figures.

5. The following statement: "...larger studies with whole genome and whole exome analyses where additional variants were mostly considered as passenger mutations." is overstated. While it is true that the greater number of variants detected by exome and/or genome sequencing in other studies have been generally used to define clonal architecture with greater precision, most publications are careful to point out that biological significance (ie, passenger vs. driver) requires either functional analysis or a demonstration of recurrence (or both). The sentence is sufficient with the clause "...where additional variants..." omitted.

We made the requested modification.